# Broadband Generation of Polarization-Immune Cloaking via a Hybrid Phase-Change Metasurface

**Ximin Tian** [1], **Junwei Xu** [1], **Ting-Hui Xiao** [2], **Pei Ding** [1], **Kun Xu** [1], **Yinxiao Du** [1,*] and **Zhi-Yuan Li** [3]

1   School of Materials Science and Engineering, Zhengzhou University of Aeronautics, Zhengzhou 450046, China; xmtian007@zua.edu.cn (X.T.); xujunwei001@zua.edu.cn (J.X.); dingpei@zua.edu.cn (P.D.); kunxu@zua.edu.cn (K.X.)
2   Department of Chemistry, University of Tokyo, Tokyo 113-0033, Japan; xiaoth@chem.s.u-tokyo.ac.jp
3   College of Physics and Optoelectronics, South China University of Technology, Guangzhou 510640, China; phzyli@scut.edu.cn
*   Correspondence: duyinxiao@zua.edu.cn

**Abstract:** Metasurface-enabled cloaking offers an alternative platform to render scatterers of arbitrary shapes indiscernible. However, specific propagation phases generated by the constituent elements for cloaking are usually valid for a single or few states of polarization (SOP), imposing serious restrictions on their applications in broadband and spin-states manipulation. Moreover, the functionality of a conventional metasurface cloak is locked once fabricated due to the absence of active elements. Here, we propose a hybrid phase-change metasurface carpet cloak consisting of coupled phase-shift elements setting on novel phase-change material of $Ge_2Sb_2Se_4Te_1$ (GSST). By elaborately arranging meta-atoms at either 0 or 90 degrees on the external surface of the hidden targets, the wavefront of its scattered lights can be thoroughly rebuilt for arbitrary SOP exactly as if the incidence is reflected by a flat ground, ensuring the targets' escape from polarization-scanning detections. Furthermore, the robustness of phase dispersion of meta-atoms endows the metasurface cloak wideband indiscernibility ranging from 7.55 to 8.35 μm and tolerated incident angles at least within ±25°. By reversibly switching of the phase states of $Ge_2Sb_2Se_4Te_1$, the stealth function of our design can be turned on and off. The generality of our approach will provide a straightforward platform for polarization-immune cloaking, and may find potential applications in various fields such as electromagnetic camouflage and illusion and so forth.

**Keywords:** metasurface carpet cloak; states of polarization (SOP); hybrid phase-change metasurface; polarization-insensitive cloaking; electromagnetic camouflage and illusion

## 1. Introduction

The electromagnetic (EM) invisibility, by eliminating the scattering light and restoring the polarization, amplitude and phase profiles of the transmitted/reflected light as if the incidence is reflected by a flat ground, can render the hidden targets unobservable, which is a dream that people have been chasing unremittingly for a long time. Such scenarios, often considered to only exist in magical movies by exploiting a specific invisible technique, such as in the Harry Potter movies, have become a reality and have flourished because of the advent of metamaterials [1,2]. Transformation optics (TO) [3], as an ambitious approach, can reboot the EM waves around the hidden objects and thereby enable it to be fully indiscernible. However, the implementation of this technique is supported by complicated constitutive parameters, especially for the extremely inhomogeneous and anisotropic profiles of the material parameters [4–6], which impose enormous challenges in practice. Afterward, the concept of carpet cloak based on scattering cancelation was proposed [7,8], which is fulfilled by suppressing the dominant multipolar scattering orders. Nevertheless, bulky footprint and lateral shift of the scattered waves can easily deteriorate the quality of cloaking and thus render the hidden objects readily detectable, especially at high frequencies. As an alternative, the cloak technique assisted by quasi-conformal mapping was also

proposed [9,10], which adopts the strategy of inverse transformation of the permittivity and permeability to release the inherent restraint of full stealth technique. However, it is generally bulky, costly and time consuming to manufacture with high precision.

Recently, the emergence of metasurfaces [11–16] undoubtedly provides a powerful platform to achieve stealth, due to its inherent virtues such as more degrees of flexible-design freedom, skin thickness, low loss and easy fabrication relative to three-dimensional (3D) bulky metamaterials. Metasurfaces, as the equivalent 2D counterpart of metamaterials, consist of subwavelength optical scatterers that could arbitrarily tailor the polarization, phase and amplitude of EM waves by introducing abrupt phase changes across the interface. Metasurface-enabled carpet cloak [17–23] inherits all the merits of metasurfaces and any targets wrapped with it can thoroughly reconstruct the scattered lights exactly as if the input EM wave is reflected by a flat conducting ground. Following this strategy, various types including multi-wavelength [24,25], reconfigurable [26,27] and smart metasurface carpet cloaks [28–30] have been proposed and demonstrated. However, these metasurface cloaks are subjected to polarization sensitivity, i.e., they work efficiently only at a single or a few states of polarization (SOP), being easily detectable by full-polarization detection systems. Such restrictions can be released by exploiting symmetric circular ring or square-shaped nanostructures as the constituent elements of metasurfaces [19,31], but at the cost of losing a degree of freedom in the design space. On the other hand, the functionality of current metasurface cloaks is locked once fabricated due to the absence of active elements, which severely limits their wide applications. As a non-volatile optical phase change material (O-PCM), $Ge_2Sb_2Se_4Te_1$, with extremely large refractive index contrast associated with material phase transformation as well as exceptionally broadband transparency and low loss in the infrared spectral regime, uniquely empowers metasurface devices with more degrees of freedom for post-processing and thus to become active cloaks. Thereafter, a reconfigurable full-polarization metasurface cloak has been in high demand but challenging to develop until now.

Here, counterintuitively, we report an ambitious approach to construct a metasurface consisting of anisotropic Au nanoantenna pairs setting on a $Ge_2Sb_2Se_4Te_1$ spacer layer, which is capable of perfect cloaking at any SOP described by the Poincaré sphere. In contrast to [21], who realized polarization-insensitive cloaking via exploiting the cross-LP scheme, in our scheme these anisotropic nanoantenna pairs with varied dimensions are arranged at either 0 or 90 degrees (relative to the *x*-axis), allowing us to accurately implement the propagation phases to cover $2\pi$ phase shifts. There are two advantages by doing so. First, both right circularly polarized (RCP) and left circularly polarized (LCP) light will be imparted by the identical phase profiles as they interact with the designed metasurface, leading to a perfect cloaking performance upon any incident SOP since any polarized light can be decomposed into a combination of LCP and RCP light. Second, the robustness of phase dispersions of these anisotropic meta-atoms endows the metasurface cloak wideband indiscernibility and large tolerated angular range. One point should be emphasized that by reversibly switching of phase states of $Ge_2Sb_2Se_4Te_1$, the designed cloak can realize stealth switching of "ON" and "OFF" by imposing appropriate external stimuli without changing the meta-devices' structures. The generality of our approach will provide a straightforward platform for reconfigurable polarization-immune cloaking, and may find potential applications in various fields such as electromagnetic camouflage and illusion and so forth.

## 2. Principles and Structures

### 2.1. Design Principles of Polarization-Insensitive Cloaking

To construct a metasurface carpet cloak that works well for metallic bumps with arbitrary boundaries of $h(x, y)$, each constituent element should be encoded with the desired compensated phase profiles $\varphi(x, y)$ aimed at the wavelength $\lambda_0$ for a specific polarization [18,21]:

$$\varphi(x,y) = \pi - 2k_0 cos\theta(h(x,y) - g(x,y)) \tag{1}$$

where $k_0 = 2\pi/\lambda_0$ is the wave number in free space, $\theta$ is the angle of input beams, $g(x, y)$ denotes the contour of a flat ground and the additional phase $\pi$ is induced by half wave loss. Generally, one can use the function $g(x, y)$ to mimic any fictive object as an illusion cloak. For simplicity, $g(x, y)$ is set to 0 to signify a flat ground herein. Note that the compensated phase profiles $\varphi(x, y)$ are only feasible for a specific polarization to realize perfect stealth. Other SOP deteriorate the entire cloaking performance due to unwanted phase distortions, which can be overcome by employing a symmetric circular ring or square-shaped nanostructures as the meta-atoms of metasurfaces. However, this approach suffers from the loss of a degree of freedom in the design space owing to the symmetry of these meta-atoms.

To solve this issue, we counterintuitively adopt anisotropic Au nanoantenna pairs as the meta-atoms of metasurfaces, which synergies propagation phases tailored by the cells' dimensions and specific PB phases by rotating meta-atoms at either 0 or 90 degrees, similar to those of our pioneered work [32] and Chen's work [33]. To be specific, as the input spin-polarized light hits on anisotropic meta-atoms, the reflected beams could be described by the Jones vector [34]:

$$R|\sigma\rangle = \frac{r_l + r_s}{2}|\sigma\rangle + \frac{r_l - r_s}{2}exp(-j2\sigma\beta)|-\sigma\rangle \tag{2}$$

in which the parameter $\sigma$ is assigned to +1 or −1 for RCP or LCP light, and $r_l$ and $r_s$ represent the complex reflection coefficients for the linearly polarized (LP) light along the long and short axes of the nanoantenna, respectively. Each nanoantenna is rotated counter-clockwise with the angle of $\beta$ with respect to the $x$-axis. From Equation (2), we can find that the outgoing EM fields contain not only the original (co-) polarized components ($\sigma$) with a complex amplitude $(r_l + r_s)/2$, but also the orthogonal (cross-) polarized components ($-\sigma$) with a complex amplitude $(r_l - r_s)/2$. It should be noted that an additional phase delay $2\sigma\beta$ is induced for the cross-polarized components, which is the origin of the PB phase. We can eliminate the unfavorable scatterings caused by the co-polarized components via optimizing the unit cells to be a perfect half-wave plates ($r_l = -r_s$), while maximizing the cross-polarized components to boost the polarization conversion efficiency (PCE) and realize high-efficiency meta-devices. However, since the PB phase is susceptible to the helicity of the incident beams, i.e., $exp(j2\beta)$ for RCP and $exp(-j2\beta)$ for LCP lights, respectively, the above cross-polarization-engineered phases are only feasible for a specific polarization, while collapsing for others. Notably, if half of all the nanoantennas are arranged at $\beta = 0°$ and the other half at $\beta = 90°$, the exponential expression results will be identical, manifesting the same phase distributions for input RCP or LCP beams, which implies that the metasurface will enable perfect cloaking upon any incident SOP.

### 2.2. Structures and Methods

Figure 1a illustrates the schematic of the polarization-immune metasurface carpet cloak, from which one could witness that the metasurface design enables perfect cloaking upon all LP (transverse magnetic, TM), LP (transverse electric, TE), LP($\pi/4$), LCP and RCP incident beams. These prototypical space-variant polarization states of incident beams can be gracefully described by the Poincaré sphere (PS) [35], as shown in Figure 2a. As a proof, an ultrathin metasurface cloak is designed, which overlays a triangular bump, as exhibited in Figure 1b. The tilt angle of the triangular bump is $a = 15°$, and $h_i = (i-1/2)\,p$ represents the vertical distance from the center point of the meta-atom ($i = 1,2,3,\ldots$ , n) to the flat ground. For better cloaking, 40 meta-atoms ($n = 40$) are arranged on each side of the triangular bump. Since the tilt angles of the two sides of the bump exhibit opposite signs, the deflection angles of reflected beams are also inverted, leading to mirror-like reflection. Left panel of Figure 1c exhibits the schematic of unit cell of the metasurface cloak, in which the spacer layers of GSST and ZnS:SiO$_2$ are sandwiched by top metallic nanoantenna pairs and bottom gold ground. The GSST layer with the fixed thickness of 490 nm functions basically as active elements to tailor local circumstances, and the ZnS:SiO$_2$ layer with a fixed thickness of 150 nm acts as the protective layer to prevent GSST from being oxidized,

respectively. The top metallic nanoantenna pairs consist of two distinct anisotropic Au nanoantennas with identical height $h$ = 100 nm and lattice constant $p$ = 3000 nm. It should be emphasized that the 3D unit cell is arranged by 0 degrees herein. To break structural symmetry and generate chirality-dependent optical responses, one antenna A0 with a fixed width $a_0$ = 550 nm and length $b_0$ = 1200 nm orients obliquely with a rotation angle $\gamma$ = 20° relative to the $x$-axis. While for the other A1, its long axis is parallel to the $y$-axis, and its width is set to $a_1$ = 800 nm and length $b_1$ varies ranging from 600 to 2900 nm to realize 0~$\pi$ coverage. Additionally, Figure 1c (right panel) also displays the top views of the meta-atoms arranged by 0 (top) and 90 (bottom) degrees.

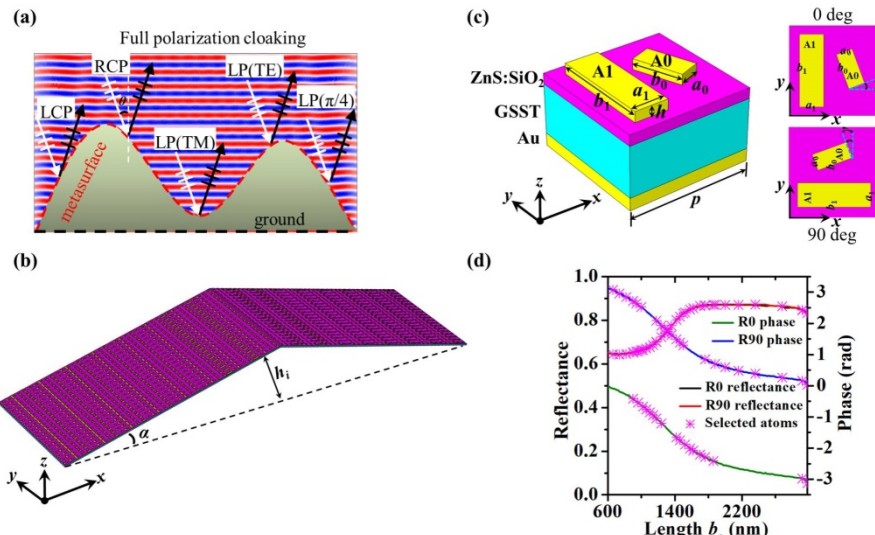

**Figure 1.** Principle diagram and design scheme of the polarization-immune metasurface carpet cloak. (**a**) Principle diagram of the polarization-immune metasurface carpet cloak wrapping on metallic bumps with arbitrary boundaries. It can work well upon all LP (TM), LP (TE), LP ($\pi/4$), LCP and RCP incident beams. (**b**) As a proof, an ultrathin triangular metasurface carpet cloak is designed in which the tilt angle of the triangular bump is $a$ = 15°, $h_i$ represents the vertical distance from the center point of the meta-atom to the flat plane. (**c**) Left: Schematic of the meta-atoms (0 degrees) of the proposed metasurface. Right: Top views of the meta-atoms arranged at 0 (top) and 90 (bottom) degrees. (**d**) Normalized reflectance and propagation phases of anisotropic Au nanoantenna pairs with length $b_1$ spanning from 600 to 2900 nm, while other parameters remain unchanged.

In the proof-of-concept demonstrations, finite-element-method-based numerical simulations are performed by utilizing the software of COMSOL Multiphysics. For each meta-atom, periodic boundary conditions (PBCs) are adopted along the $x$- and $y$-axes, and perfectly matched layer (PML) boundary conditions are implemented in the $z$ direction. The input RCP (LCP) wave propagates along the $-z$-axis and the reflection spectra and propagation phase with the flipped spin states (LCP (RCP)) are captured. While for the metasurface cloak, we set PML around the model in $x$–$z$ plane, apply PBCs along $y$-axis and adopt perfect electrical conductor (PEC) boundary instead of bottom Au ground, respectively. The incident EM waves are characterized by the incident background scattered fields that propagate along the $-z$-axis. The optical constants of GSST exhibit frequency-dependent characteristic [36], the refractive index of ZnS:SiO$_2$ is set to 2 [37] and the permittivity of Au can be represented by the Drude model [38]. A prototype of the metasurface cloak can be fulfilled by a standard lithography process, in which 490 nm GSST and 150 nm ZnS:SiO$_2$ films are DC-magnetron sputter-deposited in sequence followed by electron beam lithography, and finally 100 nm Au film is thermally evaporated.

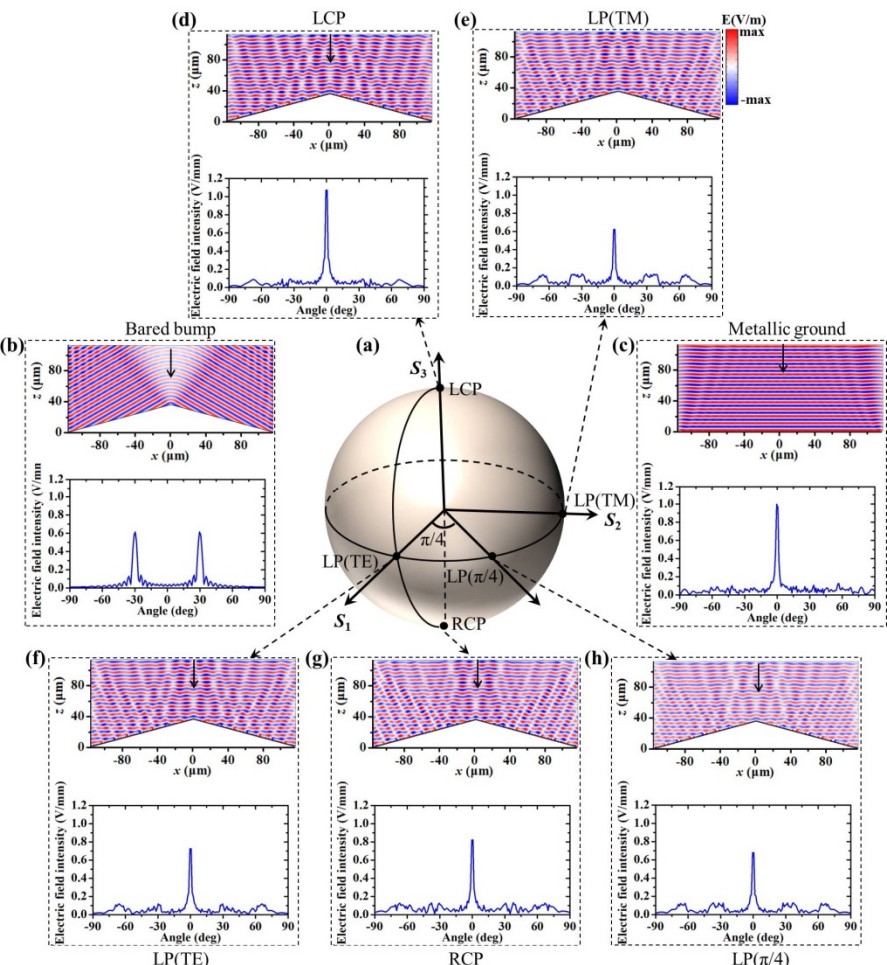

**Figure 2.** A Poincaré sphere representation of cloaking response under arbitrary SOP for normal incidence. (**a**) A Poincaré sphere and five points that characterize different SOP. The calculated reflected spatial E-field patterns in the *x–z* plane (up panels) and the corresponding farfield radiation patterns (bottom panels) for the bared bump under LCP incidence (**b**), metallic ground under LCP incidence (**c**) and the cloaked bump under LCP (**d**), LP (TM) (**e**), LP (TE) (**f**), RCP (**g**) and (**h**) LP ($\pi/4$) input light, respectively.

Figure 1d shows the simulated propagation phase shifts and reflection at the targeted wavelength of 7.9 µm by parameter sweeping of the long axis' length $b_1$ of A1 nanoantenna by varying it from 600 to 2900 nm for two perpendicular-oriented meta-atoms R0 and R90. In our simulation, R0, i.e., the meta-atoms arranged by 0 degrees, represents the long axis of A1 ($b_1$) oriented along the *y*-axis, while R90, i.e., the meta-atoms arranged by 90 degrees, represents the long axis of A1 ($b_1$) oriented along the x-axis by rotating A1 in-plane 90 degrees. It should be emphasized that the whole element structure maintains exactly the same rotation as the long axis of A1 ($b_1$) with reference to its center in the *x–y* plane. As illustrated in Figure 1d, for the meta-atoms arranged at either R0 or R90, the reflection spectra are completely coincident, while the phase shifts preserve the same profiles but are imparted with a $\pi$ phase shift, which confirms the feasibility of the theory expressed by Equation (2). As a proof, 40 units that cover $2\pi$ full-phase shifts while maintaining high reflectance are picked to build the metasurface cloak, in which 22 (close to half of 40) meta-atoms are arranged in R0 and 18 in R90, enabling the realization of polarization-insensitive cloaking performance.

### 3. Results and Discussions

With the above principle, we designed a prototype of an ultrathin metasurface carpet cloak utilizing anisotropic nanantenna pairs as building blocks and wrapping it on a triangular bump, as shown in Figure 1b. Firstly, to characterize its polarization-immune cloaking performance at normal incidence, five points that represent different SOP of incident light impinging on the designed metasurface are picked and their coordinates at the Poincaré sphere are given in Figure 2a. For the Poincaré sphere, the North (up) and South (bottom) poles denote the cases of two orthogonal circularly polarized illuminances, and Points B, C and D on the equator represent the scenarios upon LP lights with polarization angles 0 (TE), $\pi/4$ and $\pi/2$ (TM), respectively. The corresponding calculated reflected spatial E-field patterns in the *x*–*z* plane are illustrated in the up panels of Figure 2d–h, respectively. For comparison, the cases of a bared triangular bump and PEC ground upon LCP illuminance are also considered, as shown in the up panels of Figure 2b,c, respectively. Compared with the PEC ground, tangible distortions of the scattering field are induced by the bared bump, and the wavefront is substantially distributed in parallel with double tilt angle $\alpha$ (Figure 2b), very different from the case of PEC ground, for which the wavefront is parallel to the ground plane (Figure 2c). However, after wrapping the bared bump with our cleverly designed metasurface, when LCP light illuminates, scattering distortions are strongly suppressed and the wavefront is reconstructed (Figure 2d) as those of the ground plane (Figure 2c). As a consequence, the hidden target is invisible as if it does not exist. It is worth mentioning that almost identical restored wavefronts are also yielded for our cleverly designed metasurface upon all the other illuminances of RCP, LP (TM), LP (TE) and LP ($\pi/4$), implying that the metasurface cloak enables perfect cloaking upon any incident SOP. The existing little perturbation is probably due to the discontinuity of local reflection phases, which can be minimized by further optimization.

To better quantitively evaluate the quality of the generated cloaking performance by our metasurface cloak, we show the corresponding far-field radiation patterns upon the above-mentioned incident SOP, as shown in the bottom panels of Figure 2d–h. Simultaneously, the corresponding far-field radiation patterns for the bared bump and PEC ground are also depicted in the bottom panels of Figure 2b,c for comparison. As expected, two scattered sidelobes located at $\pm30°$ (double of the tilt angle $\alpha$) produced by the bared bump are efficiently suppressed and merged into one main lobe by our metasurface cloak upon any incident SOP. Such a phenomenon is very consistent with those of PEC ground, again proving that our designed metasurface enables polarization-insensitive cloaking performance at the wavelength of 7.9 µm upon normal incidence.

Furthermore, oblique incidence scenarios at 25° for the metasurface cloak upon the above-mentioned five SOP are calculated, as shown in Figure 3c–g. We also give the cases of bared cloak and PEC ground for comparison. It can be easily found that the oblique incidence of 25° breaks the symmetry of the scattered wavefront between the left and right sides for the bared bump (Figure 3a), quite different from the specular reflection of the PEC ground (Figure 3b). To our relief, after the bared bump was wrapped with the well-designed metasurface, the reflection behaviors at 25° upon all five SOP (Figure 3c–g) were analogous to those of the PEC ground, explicitly implying that the designed metasurface cloak is insensitive to the incident SOP even at modest oblique angles, and straightforwardly proving that the unique design could tolerate incident oblique angles of at least 25 degrees at the wavelength of 7.9 µm.

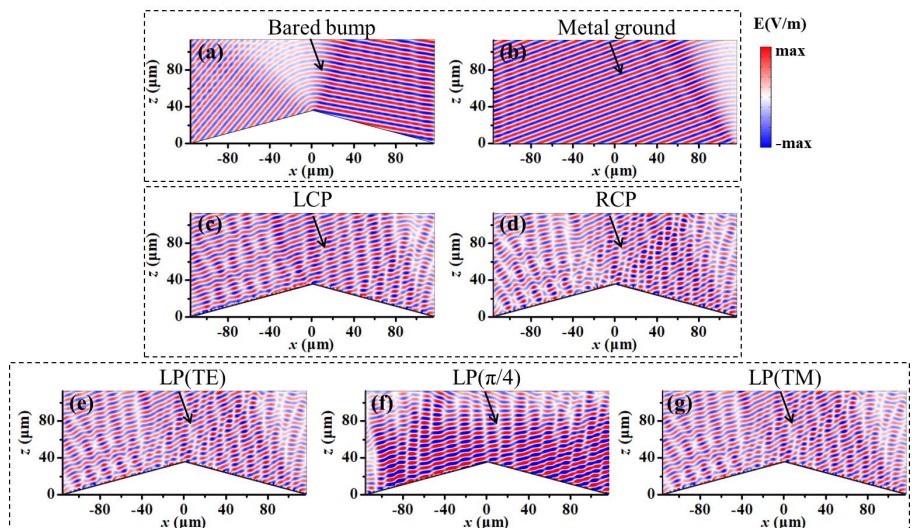

**Figure 3.** Cloaking response of our design upon arbitrary SOP under oblique incidence. For our design under the oblique incidence of 25°, the simulated reflected spatial distributions in the *x*–*z* plane for the bared bump under LCP incidence (**a**), metallic ground under LCP incidence (**b**) and the cloaked bump under LCP (**c**), RCP (**d**), LP (TE) (**e**), LP (π/4) (**f**) and LP (TM) (**g**) illuminance, respectively.

Broadband cloaking operation is a crucial indicator to evaluate the performance of invisibility cloaks. For convenience, we only showed the simulated far-field radiation profiles of the cloaked bump upon normal LCP incidence against the input wavelength from 7 to 9 μm in Figure 4a. One could see that the far-field radiation profiles are converged into a straight horizontal line located at 0 degrees within a wideband from 7.55 ($\lambda_1$) to 8.35 μm ($\lambda_2$) (labelled by dashed white lines), i.e., our metasurface cloak exhibits a broadband cloaking response despite some trivial scattering perturbations. To reveal the underlying mechanism, we calculated the theoretical phase dispersions of all the picked meta-atoms according to Equation (1) under three prototypical incident wavelengths (7.55, 7.9 and 8.35 μm) in Figure 4b. The actual (simulated) phase dispersions for all selected meta-atoms at 7.9 μm are also displayed. One could observe that the theoretical predictions exactly match with the actual phases at $\lambda_0 = 7.9$ μm, leading to invulnerable invisibility. Although the theoretical phase lines at $\lambda_1 = 7.55$ μm and $\lambda_2 = 8.35$ μm slightly deviate from the actual ones, the designed metasurface cloak can still preserve perfect invisibility due to the robustness of the phase dispersion of meta-atoms [39]. Figure 4d,e shows the simulated reflected E-field intensity profiles in the *x*–*z* plane and their corresponding 2D far-field radiation patterns for our scheme under LCP illuminance with $\lambda_1 = 7.55$ μm and $\lambda_2 = 8.35$ μm, respectively. As expected, these reconstructed wavefronts exhibit almost identical morphological features to the one at the wavelength of 7.9 μm, unanimously confirming that the metasurface can obtain broadband and invulnerable invisibility in the MIR. To further verify the broadband cloaking operation of our design, Figure 4c depicts the reduced total radar cross-section (RCS) that is calculated by dividing the total RCS of the cloaked bump by that of the bared bump [40]. It can be easily found that our design upon normal incidence produced a significant RCS reduction reaching up to −18 dB, and the 3 dB RCS reduction bandwidth is very consistent with the results in Figure 4a, further implying the feasibility of the broadband cloaking operation of the designed metasurface. Additionally, the relatively large RCS reduction upon oblique incidence (θ = 25°) demonstrates the virtues of wide-angle cloaking very well.

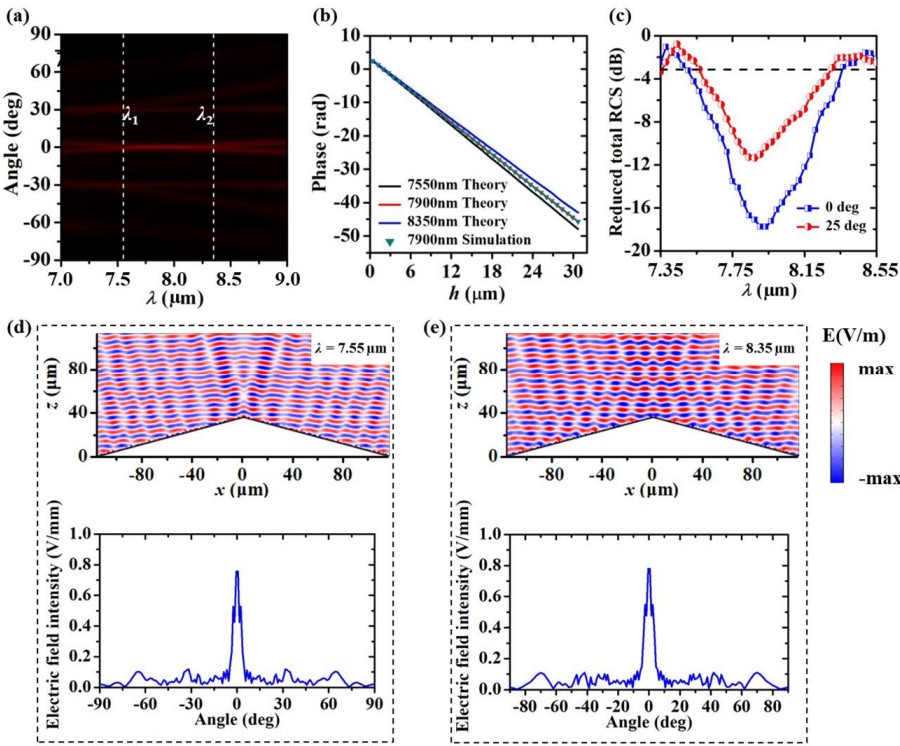

**Figure 4.** Broadband cloaking operation of the polarization-insensitive metasurface carpet cloak. (**a**) Scanned plot of the simulated far-field radiation profiles for the cloaked bump upon LCP incidence. (**b**) Phase dispersions of all picked meta-atoms under three prototypical incidences. (**c**) Calculated RCS reduction for our metasurface cloak under normal and oblique incidences. Simulated reflected E-field intensity profiles in the $x-z$ plane (up panels) and the corresponding 2D far-field radiation patterns (bottom panels) for our scheme under LCP illuminance with $\lambda_1 = 7.55$ μm (**d**) and $\lambda_2 = 8.35$ μm (**e**), respectively.

It should be emphasized that our metasurface cloak is endowed with reconfigurability and continuous (multilevel) control supported by embedding a chalcogenide phase-change layer of GSST as a spacer layer. As a representative chalcogenide PCM, GSST could be switched instantaneously, repeatedly and non-volatilely along amorphous (aGSST) intermediate and crystalline (cGSST) states [41]. A huge contrast in the complex refractive index caused by phase transformation and low loss in the MIR enable GSST to be a good platform for reconfigurable and versatile optics [36]. To confirm this fact, Figure 5 displays the simulated reflected E-field intensity profiles in the $x$–$z$ plane for our scheme with different crystallization levels ($m$ = 0.25, 0.5, 0.75 and 1) upon LCP illuminance of $\lambda_0 = 7.9$ μm, respectively. For the calculation of the effective permittivity of GSST at any crystallization level, please refer to Ref. [42]. $m$ = 0 and $m$ = 1 indicate that GSST exists in amorphous and crystalline states, respectively, and the larger the value of $m$, the higher the crystallinity of GSST. This work adopts aGSST by default, unless otherwise specified. The results of the scenario with $m$ = 0 have been shown in Figure 2d. As anticipated, the stealth performance of the designed metasurface cloak gradually deteriorates as $m$ increases, which is mainly attributed to the red shift of resonance and enlarged loss. When $m$ = 1 (Figure 5d), the reflected wavefront of our design looks very similar to that of the bared bump (Figure 2b), indicating that the designed metasurface cloak turns off. Therefore, by adjusting the crystallinity of GSST, our designed metasurface cloak can realize the function of continuous regulation and switching, which undoubtedly expands the flexibility and diversity of the designed functionalities.

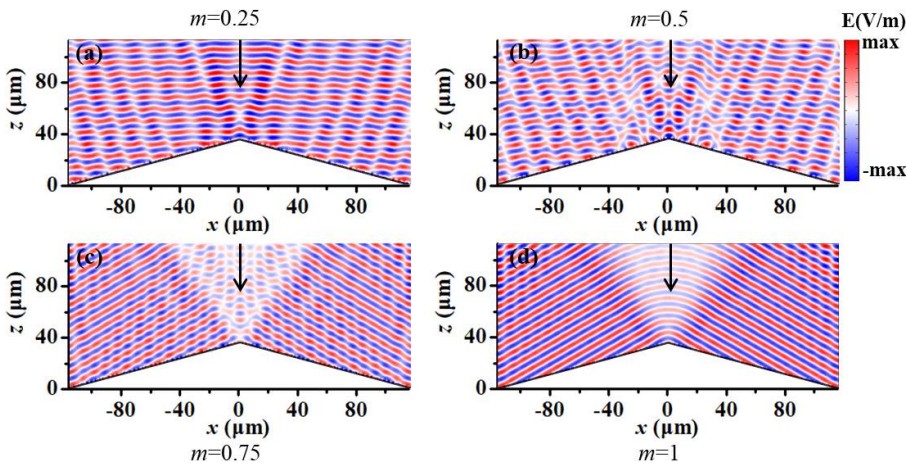

**Figure 5.** GSST state-dependent cloaking performance of our design under normal incidence. Simulated reflected E-field distributions in the x–z plane for our scheme with different crystallization levels of $m$ = 0.25 (**a**), 0.5 (**b**), 0.75 (**c**) and 1 (**d**) upon LCP illuminance with $\lambda_0$ = 7.9 μm nm, respectively.

## 4. Conclusions

In summary, we report a $Ge_2Sb_2Se_4Te_1$-based metasurface carpet cloak consisting of 40 anisotropic nanostructures. By arranging well-optimized meta-atoms at either 0 or 90 degrees on the external surface of the hidden targets (triangular bared bump), $2\pi$ full-phase coverage can be obtained, ensuring the wavefront of scattered lights can be thoroughly rebuilt exactly as if the incidence is reflected by a flat ground under arbitrary SOP of the Poincaré sphere, and thus enabling the hidden targets to escape from polarization-scanning detections. Meanwhile, the robustness of the phase dispersion of meta-atoms allows for the metasurface cloak to generate a broadband indiscernibility within the wavelength range of 7.55 to 8.35 μm and to tolerate the incident angles reaching up to $\pm25°$. It should be emphasized that our metasurface cloak is endowed with a continuously adjustable and switchable stealth response supported by converting GSST states. The generality of our approach will provide a straightforward platform for polarization-immune cloaking, and may find potential applications in various fields such as electromagnetic camouflage and illusion and so forth.

**Author Contributions:** Conceptualization, X.T. and J.X.; methodology, X.T.; software, X.T. and J.X.; formal analysis, X.T. and J.X.; investigation, X.T., J.X. and T.-H.X.; data curation, X.T. and J.X.; writing—original draft preparation, X.T. and J.X.; writing—review and editing, X.T., J.X., K.X. and Y.D.; supervision, X.T., Y.D. and Z.-Y.L.; project administration, X.T., J.X., Y.D. and P.D. All authors have read and agreed to the published version of the manuscript.

**Funding:** This research was funded by the National Natural Science Foundation of China (No. 12004347); the Scientific and Technological Project in Henan Province (Nos. 212102310255, 202102310535, 222102210063); the Aeronautical Science Foundation of China (Nos. 2020Z073055002, 2019ZF055002); and the Innovative Research Team (in Science and Technology) in the University of Henan Province (No. 22IRTSTHN004).

**Institutional Review Board Statement:** Not applicable.

**Informed Consent Statement:** Not applicable.

**Data Availability Statement:** The data presented in this study are available upon request from the corresponding author. The data are not publicly available because the data also forms part of an ongoing study.

**Acknowledgments:** The authors express their appreciation to the anonymous reviewers for their valuable suggestions.

**Conflicts of Interest:** The authors declare no conflict of interest.

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
