# Peer review of "Broadband Generation of Polarization-Immune Cloaking via a Hybrid Phase-Change Metasurface"

_photonics, doi:10.3390/photonics9030156_

Round 1

Reviewer 1 Report

In the present paper, the authors extend their previous work [25] and provide a metasurface cloak that is insensitive to the polarization of the impinging wave. The analyses and results seem sound and interesting, however there are certain points that need to be addressed before any decision on the publication of the paper can be made.

I did not quite understand what the novelty of the paper is compared to similar studies, such as Ref. 21 where the authors similarly use a anisotropic meta-units to achieve polarization insensitive metasurface-based cloaking. The authors should be explicit about what novelties they are bringing with their study.

I understand that the authors envisage to use phase-change materials to enable some kind of configurability. But for me this seems very trivial. If one were to change the material properties of a cloak, its cloaking performance would obviously deteriorate. I feel that the authors somewhat overstate this basic fact by making overhyped statements such as "By reversibly switching of the phase states of Ge2Sb2Se4Te1, we can render the stealth function of our design normal or invalid". Obviously, if you change the phase profile the cloaking will become "invalid". There is nothing surprising about that!

There is not enough information between Eq. (2) and the design shown in 1(c). How did the authors come up with this design, i.e. how did they envisage this to be polarization-insensitive. Are there any general design rules for that?

Also the authors use a somewhat "weird" language. For example:

"frequecy-reliant characteristic" Nobody uses the term "frequency-reliant". It's typically "frequency-dependent".

"metasurface cloak turns invalid." Again nobody says that the cloak turns "invalid". It's typically "the cloak turns off"

There are many examples like these, they need to be corrected according to the canonical usage of terms in the literature.

Reviewer 2 Report

The article is carefully written and represents an interestingly novel approach to the applications of metasurfaces. I would like the authors to make it more clear if they see an important distinction between what they refer to as a carpet cloak and what they refer to as mantle cloak. If no distinction is intended, the authors could consider using a single term consistently. Also, the choice of the exotic material that the cloak uses could be motivated in qualitative terms in the introduction, especially because the material is novel. Apart from these minor remarks, the article seems precise in its scientific content and deserves to be published after minor changes.

Reviewer 3 Report

In this manuscript, the authors  proposed an ingenious approach which allows to realize a polarization insensitive cloaking.  Recent advances in materials provide an increasing interest for metasurfaces engineering  development.This is especially true for manipulation of polarization states that are capable of being applied in imaging processing, information technologies and electromagnetic illusions. In this paper  a Ge2Sb2Se4Te1-based prototype of ultrathin metasurface carpet cloak consisting of 40 anisotropic nanostructures was performed. Authors propose realistic designs that may realize this concept. The introduction is very well written, the main relevant references under the subject are included. The design and simulation of the proposed metasurface described in this manuscript are rigorously performed, and the conclusions are well presented and clearly described. The work should be accepted for publication in Photonics after minor revisions:

  1. It is necessary to rewrite repeated sentences in the annotations and in the introduction, for example  “ We envision that …”
  2. Add a reference to the article “Mantle cloaking using thin patterned metasurfaces” (Phys Rev B, 2011) in the introduction section.
  3. Remake the figure 1(c), because it is completely unreadable. Almost invisible a rotation angle gamma, etc.
  4. It is necessary to replace the wavelength range “from 7550 nm to 8350 nm”  to “from 7,55 um to 8,35 um”.
  5. Why do authors choose this wavelength range? What is the practical application of such a structure in the range of 7,55-8,35 um ? 
  6. The manuscript contains a number of minor grammatical errors in its use of English.

Round 2

Reviewer 1 Report

The authors have provided adequate responses to the concerns raised by the reviewers. However, I would loved to see some additions/revisions to the manuscript based on the first two points I have revised (rather than just a response). Overall, the paper may warrant publication in Photonics even though I do not think that the novelty and impact of the work is extremely high.
